# Bibliometric Analysis of the Scientific Production on Compassion Fatigue

**DOI:** 10.3390/jpm12101574

**Published:** 2022-09-24

**Authors:** Luís Sousa, Bruno Ferreira, Paulo Silva, Margarida Tomás, Helena José, Esperanza Begoña Garcia-Navarro, Ángela Ortega-Galán

**Affiliations:** 1Escola Superior de Saúde Atlântica, 2730-036 Barcarena, Portugal; 2Comprehensive Health Research Centre (CHRC), 7000-811 Evora, Portugal; 3Hospital Beatriz Ângelo, EPE, 2674-514 Loures, Portugal; 4Unidade Local de Saúde do Baixo Alentejo, EPE, 7801-849 Beja, Portugal; 5Campus El Carmen, Universidad de Huelva, Avda. Tres de Marzo, s/n, 21071 Huelva, Spain

**Keywords:** compassion fatigue, compassion, bibliometrics

## Abstract

Background: Compassion fatigue is a common phenomenon among healthcare professionals and includes several concepts that share a direct relationship with quality of life, with consequences on both physical and emotional well-being but also at the economic and organizational levels. Objectives: To analyze the profile of scientific publications on compassion fatigue, dissecting trends, and highlighting research opportunities. Method: Bibliometric analysis based on Donthu’s guidelines, data collection from Web of Science (Clarivate Analytics), and analytic techniques (performance analysis and science mapping) with VOSviewer^®^ and CiteSpace^®^. Results: We obtained 1364 articles and found that the concept emerged in 1995 and is frequently associated with areas of general health. Through analysis, we identified the following research frontiers: “vicarious traumatization”, “working”, “survivor”, “mental health”, and “impact”. Conclusion: There has been a growing interest in this subject among researchers, with an increase in scientific production related to areas of health such as nursing, providing a solid starting point for further investigation. Registration number from the Open Science Framework: osf.io/b3du8.

## 1. Introduction

Compassion fatigue is a common phenomenon in healthcare professionals conceptualized as emotional, physical, and psychological exhaustion and dysfunction related to the exposure to chronic work-related stress and compassion stress [1,2,3].

Compassion materializes as a feeling derived from witnessing people’s experiences of suffering, constituting the underlying motivation toward helping others [4]. However, the nature of health professionals’ work, entailing an everyday delivery of care, means the possibility of experiencing compassion fatigue. Insufficient awareness or training in identifying its signs and symptoms in themselves or in colleagues allows the problem to develop [5].

The most affected by this phenomenon are nurses, who are also the largest group of healthcare providers. Intrinsic to nursing practice is the support of patients in their physical, mental, emotional, and spiritual needs in an empathic manner, which entails a stressful and high-risk profession, thus resulting in an increase in vulnerability to symptoms of compassion fatigue [1,2].

Thus, compassion fatigue encompasses an assortment of concepts at its core, all directly linked with the quality of life. Amongst health professionals, nurses have a relevant role in health care, for it is expected that they provide high-quality, person-centered care in an empathetic and compassionate way, bringing to realization the purpose of their care of increasing the quality of life, which, in turn, entails a need for a sufficient quality of life to perform their care [6]. 

Many professionals in the daily provision of care may suffer from compassion fatigue and may not be alert or have the knowledge to identify the signs in themselves or in their colleagues [5].

A meta-analysis that included 79 studies involving 28,509 nurses from 11 countries found the levels of compassion satisfaction and compassion fatigue among nurses to be moderate. Furthermore, the findings revealed that nurses in the Asian region working in intensive care units suffer from severe symptoms of compassion fatigue, and the prevalence of compassion fatigue has been increasing over time [1].

Another meta-analysis that included 21 studies involving 6533 oncology nurses from six different countries revealed a “moderate” level of satisfaction with compassion, burnout, and secondary traumatic stress, and that 22% of these nurses suffered from a “high” risk of compassion fatigue [3].

Recent evidence on the issue of compassion fatigue among nurses has revealed serious consequences for their physical, psychological, and spiritual health. In this context, compassion fatigue is associated with insomnia, exhaustion, depression, lower job satisfaction, loss of hope, lack of nutrition, lack of spiritual awareness, and poor judgment [1,7,8]. When the levels of compassion fatigue in nurses increase, the quality of the nurse–patient relationship worsens, the relationship becomes more strained [9], conflicts in the health team increase, and there is less sympathy and empathy at work [8]. At the organizational level, compassion fatigue is negatively associated with organizational outcomes, with increased costs for healthcare institutions, reduced productivity, and high staff turnover [1,10]. Given the relevance of the previous studies’ findings, it is crucial to know the scientific production associated with compassion fatigue to verify trends, co-citation, and research boundaries in order to take measures to reduce compassion fatigue and, therefore, the consequences concerning nurses’ health, their relationship with their patients, and their performance within health organizations.

This research was guided by the following question: What are the global trends in the scientific evidence on compassion fatigue? The bibliometric analysis aims to analyze the profile of scientific publications on compassion fatigue and their trends and highlights research opportunities.

## 2. Materials and Methods

### 2.1. Type of Research

Bibliometrics is a research method that enables the quantitative measurement of the results of bibliographic research using specific statistical methods [11].

In the analytical scope, the use of mathematical and statistical tools allows the distribution of information regarding publication and communication patterns [12].

This methodological approach makes it possible to analyze the metric characteristics of the data obtained in a given research domain, considering a predefined temporal spectrum [12,13,14,15].

The results obtained through the application of statistical methods enable a qualitative analysis, which facilitates the understanding of the domain’s progress under analysis and highlights emerging trends in the performance of scientific production items [16]. This qualitative approach makes it possible to assess the contribution of knowledge derived from scientific evidence produced in a given conceptual area, identifying opportunities for further research [12,16].

Hence, this is a mixed-methods study [14] which allows the identification of research boundaries of the domain under analysis, highlighting related terms that deserve to be developed [16].

Recent guidelines for this type of study have guided the performed analysis, which consists of four steps: (1) a definition of the study objective; (2) the choice of the analysis technique; (3) the collection of data for analysis; and (4) the performance of the bibliometric analysis and exposing its results [16].

### 2.2. Research Strategy, Study Selection, and Data Extraction

To ensure that there are no bibliometric studies in the field of compassion fatigue, we carried out a preliminary search on the Web of Science (Clarivate Analytics), Scopus (Elsevier), and EBSCO (all databases). No bibliometric studies were identified in this scope, so we executed a subsequent search in the Virtual Health Library (VHL), and the following DeCS/MeSH descriptor was found: “Compassion Fatigue” with no alternative terms identified in that same search. Regarding evidence selection criteria, we included reviews and empirical studies published in scientific journals in article format addressing or referring to the concept of compassion fatigue, with no chronological limit of publication, and published in any language. A search in Web of Science (Clarivate Analytics) was then performed with the following search equation: AK = (“Compassion Fatigue”) OR KP = (“Compassion Fatigue”), refined by: Document Types: (ARTICLE OR REVIEW ARTICLE). Indexes: SCI-EXPANDED, SSCI, A&HCI, CPCI-S, CPCI-SSH, ESCI, CCR-EXPANDED, IC.

The search was performed on 28 June 2022 at 18:05 GMT + 1 in order to avoid bias arising from the constant updating of the databases. After the search in the Web of Science (Clarivate™), data collection was conducted in a single day, carried out with a file compatible with the bibliometric analysis software used: CiteSpace^®^ 5.7.R2 (Drexel University, Philadelphia, PA, USA) and VOSviewer^®^ (Leiden University, The Netherlands).

The search and data extraction were validated by two independent researchers, and, in occasional cases of any doubt or inconsistency in the results, these were analyzed by a third researcher and discussed by the research team in order to obtain consensus.

### 2.3. Statistical Analysis of the Data

The statistical analysis process was carried out simultaneously with the extraction of bibliometric data, following the most recent guidelines. This process consisted of the development of two main techniques: performance analysis, which examines the contributions of research constituents, and scientific mapping, which examines the intellectual interaction and structural links between the constituents of the research [16].

The description of the data is based on graphics, figures, and tables that were produced by analysis software such as VOSviewer^®^ [15], CiteSpace^®^ [17], and Microsoft Excel^®^.

## 3. Results

### 3.1. Trend in the Annual Evolution of Publications

From a sample of 1364 published articles obtained since 1995, as shown in Figure 1, we can see a consistent increase in the number of publications since 2009, with a higher publication number in 2020.

### 3.2. Distributions of Articles by Journal and Area of Research

Of the 74 research areas found, the 5 with the highest number of publications are: Nursing (n = 424), corresponding to 31,085% of all the published articles, Psychology (n = 271), corresponding to 19,868%, Psychiatry (n = 153), corresponding to 11,217%, Social Work (n = 148), corresponding to 10,850%, and Occupational Health (n = 110), corresponding to 8065%.

Regarding the publication titles, it is possible to observe (Table 1) the 5 with the highest impact factor out of a universe of 583. From 1364 articles, *International Journal of Environmental Research* is responsible for 26 publications (1.906%), *Journal of Advanced Nursing* for 24 (1.760%), *Journal of Nursing Management* for 22 (1.613%), *Frontiers in Psychology* for 20 (1.466%), and *Journal of Clinical Nursing* for 17 (1.246%). Taking into account the five journal titles with more publications, it is possible to observe that the impact factor, according to Journal Citation Reports (Clarivate™), is ranked from 2.988 to 3.390. According to the Scimago Journal & Country Rank, these same publication titles are ranked from 0.81 to 1.11, corresponding, at the time of the search, to quartile one.

### 3.3. Distribution of Articles by Language and Country of Publication

Of the 72 countries with published articles regarding this topic, it is possible to highlight in Figure 2 that the majority of articles have their origin in the United States of America, with 593 published articles, corresponding to 43.475% of all publications, followed by Australia, Canada, the United Kingdom, and Spain, with 124 (9.091%), 108 (7.918%), 101 (7.404%), and 62 (4.545%), respectively. Out of a total of 1364 articles, the vast majority are written in English (97.801%), which corresponds to 1334 articles. The other 30 articles are written in nine different languages. 

When analyzing the Citespace^®^ information regarding the clusters from which the publications originated, it is possible to see 77 nodes referring to countries and 217 connections between them. Five clusters have appeared with sizes between 5 and 22, with a silhouette index between 0.757 and 0.888 with an average year of 2014. Figure 3 shows the three countries with the strongest citation bursts.

### 3.4. Author Profile

From the 4402 collaborations, 1364 articles were produced, highlighting the 10 authors with the highest number of published articles, with Galiana and Sanso being the authors with the most publications, followed by Hegney, Oliver, Rees, Ortega-Galán, and Craigie, as verified in Figure 4. 

When analyzing the authorship clusters on Citespace^®^ (Figure 5), we observed 221 authorship nodes with 217 connections between them, highlighting only one cluster with a size of 15 and a silhouette index of 1 with an average year of 2014, consisting of five articles [18,19,20,21,22].

Figure 6 shows the top 10 authors with the strongest citation bursts.

We analyzed the co-authorship density with VOSViewer^®^. Figure 7 shows this density of co-authorship with 29 authors distributed by five clusters with 130 connections between them.

### 3.5. Article Profiles 

Out of a total of 1364 articles included in the research, 1223 are original articles, corresponding to 89.663%, and 141 are review articles, corresponding to 10.337%.

Regarding the citation profile, out of a total of 22,242 citations, the three most cited studies have 606 citations (2.725%) [23], 374 citations (1.682%) [24], and 314 citations (1.412%) [25], with 2020 being the year in which these three articles had the highest number of citations. With the information from CiteSpace^®^, it was possible to identify 716 nodes referring to the cited articles, with 52,199 links between them, highlighting four clusters with sizes between 15 and 265 and a silhouette index between 0.634 and 0.915 with an average year of 2008. Figure 8 shows the top five references with the strongest citation bursts.

Concerning the term co-occurrence in titles and abstracts, and analyzing the clusters from VOSViewer^®^, there were found to exist 100 terms divided into four clusters with 4648 connections between them. Secondary studies were observed in early 2017, and words such as “article, literature, review and concept” are currently appearing. From 2019, terms such as “control group, cross-sectional study” start to emerge, as shown in Figure 9. 

Analyzing the co-occurrence of keywords, and analyzing the information from Citespace^®^, we found the existence of 452 nodes referring to keywords used in articles, with 10,693 links between them and six clusters standing out with a size between 9 and 113 and a silhouette index between 0.461 and 0.935, as seen in Figure 9.

When analyzing the Citespace^®^ information regarding the keywords with the strongest citation bursts, the five keywords with the strongest citation bursts were found to be “Vicarious traumatization, Working, Survivor, Mental Health and Impact”, all of them with citation bursts from 1995 onwards, but only “Vicarious traumatization, Working and Survivor” with citation bursts remaining until today. Those three keywords are also the ones with the strongest bursts of citations, as verified in Figure 10. It is also possible to identify 452 nodes referring to keywords with 10693 links between them. These six clusters have a size between 9 and 113, with a silhouette index between 0.528 and 0.935 with an average year of 2014.

## 4. Discussion

A bibliometric analysis of global trends in the research on the concept of compassion fatigue was conducted. The timeframe was established from the first publication on the topic (1995) to the present (2022).

The analysis shows a constant increase in the number of publications over the years until 2020, with a decrease in publications from 2020 to 2021. This fact may be due to the pandemic, since most journals prioritized topics related to COVID-19.

English is the most widely used language, and the United States of America is the country that has contributed the most to research on this subject, followed by Australia, Canada, the United Kingdom, and Spain.

Of the 583 scientific journals that published on this topic, the five journals with the highest number of publications were the following: *International Journal of Environmental Research* (26 publications), *Journal of Advanced Nursing* (24 publications), *Journal of Nursing Management* (22 publications), *Frontiers in Psychology* (20 publications), and *Journal of Clinical Nursing* (17 publications), with impact factors, according to Journal Citation Reports (Clarivate™), of 3.390, 3.187, 3.325, 2.988, and 3.036, respectively.

Nursing was the area where the most research was published on compassion fatigue, corresponding to 31% of the articles of the total sample analyzed, followed by Psychology with 19.868%, and Psychiatry with 11.217%. As the phenomenon under study is very common among nurses, this professional group has been described as the most vulnerable in the health sector [1,2]; accordingly, the number of publications has been greater in the field of nursing.

The authors who produced the most articles on the subject were Galiana and Sanso, followed by Hegney, Oliver, Rees, Ortega-Galán, and Craigie. Furthermore, when analyzing the authorship clusters and respective links, we found there was a cluster of production from 2014 onwards consisting of five articles [18,19,20,21,22].

The most cited study, “Compassion fatigue: Psychotherapists’ chronic lack of self-care” [23], has 606 citations to the date of this analysis. The analysis of the co-occurrence of citations shows nursing and fatigue with greater expression. In the last two years, there has also been an emergence of keywords such as COVID 19, pandemic, control group, and mindfulness. These data show that the COVID 19 pandemic is associated with compassion fatigue [26,27,28,29,30] and the appearance of experimental studies [31,32,33] with mindfulness programs [31,32,33,34,35,36].

### 4.1. Research Frontiers

The five keywords that had the strongest citation bursts were “Vicarious Trauma”, “Work”, “Survivor”, “Mental Health”, and “Impact”, starting in 1995; however, the citations bursts “Mental Health” and “Impact” finished in 2014.

#### 4.1.1. Compassion Fatigue

Compassion fatigue is defined as a decreased ability or interest in supporting the suffering of others [23]. This phenomenon has been characterized by symptoms such as apathy, depression, anxiety, feelings of helplessness, anger, clinical errors in judgment, intrusive thoughts, sleep disturbances, and hypertension [37].

In a study on the dynamic properties of citation flows [38], it was found that in the first years after the publication of articles there are bursts of citations that can be considered as an indicator of the popularity of the evidence produced. It is well known that the popularity of the articles may show a different durability over time.

The term compassion fatigue is associated with burnout, post-traumatic stress, empathy, and compassion satisfaction [39], which may in general have conceptual framing difficulties. 

In more recent studies, this concept appears associated with the concepts of compassion satisfaction, burnout [30,37,40,41,42], quality of life [29,43,44,45,46], and resilience [47,48]. As with the co-occurrence of more recent keywords, experimental studies are beginning to emerge, in which strategies such as mindfulness [31,33,34] are used to reduce or improve coping with compassion fatigue. The COVID-19 pandemic also boosted studies on this concept [27,30].

#### 4.1.2. Work

Studies on compassion fatigue present the concepts of “working” or “work”, thus reflecting the work context. Compassion fatigue is common in certain work contexts, specifically in healthcare and social work. The most recent studies included in this analysis are mostly carried out with nurses [35,36,41,42,45,46,47,49] and in contexts where the work of nurses is more vulnerable and susceptible to compassion fatigue, such as cancer care [50], pediatric oncology care [51], palliative care [28,32,43], and the most recent context, the COVID-19 pandemic [26]. In addition, a qualitative study has emerged in the context of social work [52]. 

Health professionals are susceptible to developing compassion fatigue, in which nurses stand out because they are the largest professional group and because of their functional content [1,2].

Furthermore, in addition to the individual consequences that compassion fatigue has on health professionals, especially nurses, there are other consequences at the organizational level. Due to the retention rate of nurses in organizations, there are immeasurable costs incurred, and healthcare organizations cannot downplay this situation in a global environment with critical nursing shortages [53].

#### 4.1.3. Vicarious Trauma and Survivor

Vicarious trauma has been defined in many ways and has been given many different names, such as secondary victimization, contact victimization, compassion fatigue, and secondary traumatic stress [54].

The keywords “vicarious trauma” and “survivor”, which are included in the top five keywords with stronger citation bursts, can be associated with one another. There seems to be a mediating role of survivor guilt between empathy and compassion fatigue in nurses [55]. However, these keywords have been used less, and, in this research, they appear in few studies [55,56,57]. Considering the above, there is a need to clarify this concept through an evolutionary concept analysis.

### 4.2. Implications for Practice

Given the increasing prevalence of this phenomenon in the health sector and in particular among nurses, measures must be taken to prevent and minimize the consequences of compassion fatigue, namely, physical, psychological, and organizational measures, through a correct assessment as well as the development and implementation of intervention programs aimed at the most vulnerable health professionals and caregivers [40,51].

Due to nurses representing the largest number of elements in the health system, and due to their functional content, they have the highest rates of compassion fatigue.

### 4.3. Strengths and Limitations of the Study

In this bibliometric analysis on compassion fatigue, we were able to identify the scientific production on this topic, the scientific areas, the main authors, the most relevant articles, the co-citations, and the articles with the greatest bursts of citations. We were also able to identify associated keywords, as well as search boundaries and trends.

This bibliometric analysis has some limitations. Firstly, the number of articles analyzed and extracted was quite large, leading to an exhaustive treatment, but the software had limitations in the treatment of large amounts of data. Therefore, some publications by reference authors may not have been analyzed because they were not cataloged correctly or were not cataloged according to the inclusion criteria. Next, other types of publications, such as conference abstracts or books, were not included. Moreover, articles published in 2021 and 2022 may not have been indexed, since in some journals this process takes more than six months, and therefore they were not included in the analysis, which may have had some impact on the final result. Finally, another limitation may be that only Web of Science was used, excluding SCOPUS and EBSCO among others, since they have different ways of cataloging the metadata, which prevents a joint analysis by the software used.

### 4.4. Future Indications

Although it seems to us that the results found in this bibliometric analysis are valuable for clinical practice, teaching, and research, we still see a need to develop more studies to complement the analysis, namely, more influential universities researching on compassion fatigue and a systematic review with a meta-analysis on the interventions that have the best results in reducing compassion fatigue among nurses.

Regarding the conceptualization of the term, there are authors who have discussed the adequacy of the concept “compassion fatigue”, as it assumes that compassion and its attributes are the cause of fatigue in professionals. In the future, a new conceptual development of the term applied to the field of health sciences could be proposed [45]. Given this lexometric heterogeneity, and to avoid ambiguity in the applicability of the concept of compassion fatigue, it will be important to clarify it in a future study, using the Rodgers’ method [58].

## 5. Conclusions

With the production of this article, it was possible to conclude that there has been a growing interest in this subject among researchers, with the increase in the scientific production one of the main justifications for it. It is also plausible to say that the subject is intimately related to work and the areas of general health, particularly nursing.

This subject is still recent, appearing in 1995, and may benefit from an evolutionary concept analysis based on the Rodgers’ framework for the clarification of the concept, potentially contributing to its evaluation and intervention and having implications for practice. 

Regarding the investigation itself, it is possible to conclude that the subject requires further investigation as it is a subject with implications for today in the nursing context which may be incorporated into future research on the subject, providing a solid starting point for further investigation. 

## Figures and Tables

**Figure 1 jpm-12-01574-f001:**
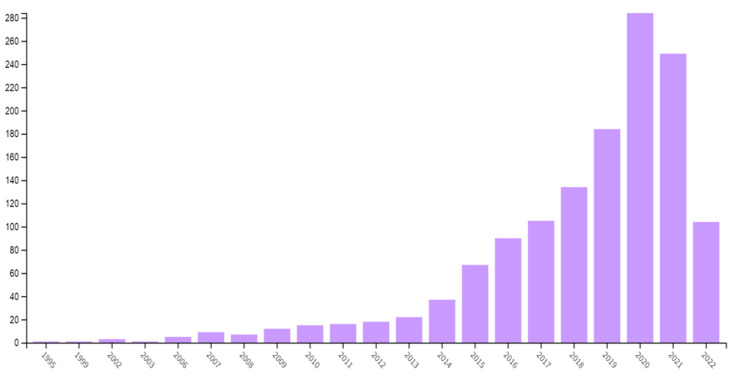
Number of publications by year.

**Figure 2 jpm-12-01574-f002:**
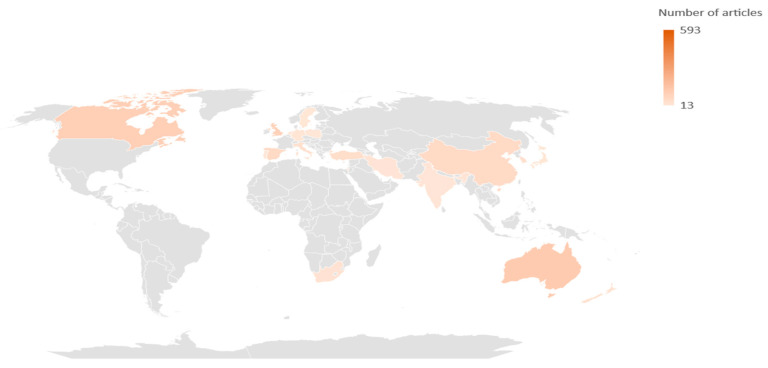
Distribution of articles by country.

**Figure 3 jpm-12-01574-f003:**
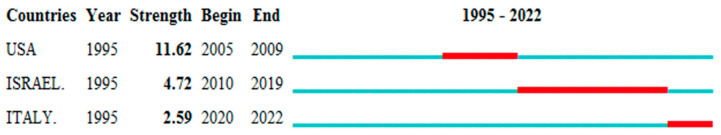
Top 3 countries with strongest citation bursts.

**Figure 4 jpm-12-01574-f004:**
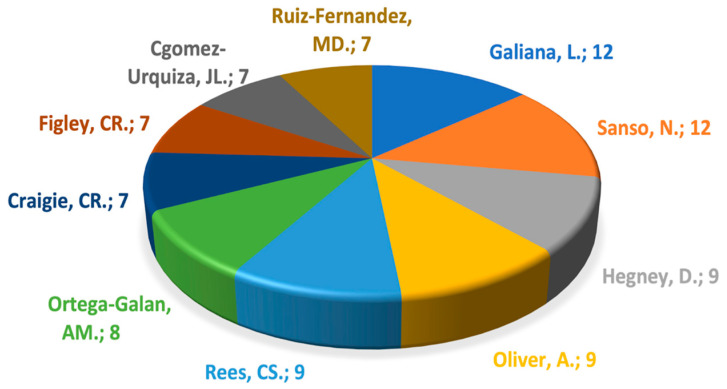
Top 10 authors with highest number of published articles.

**Figure 5 jpm-12-01574-f005:**
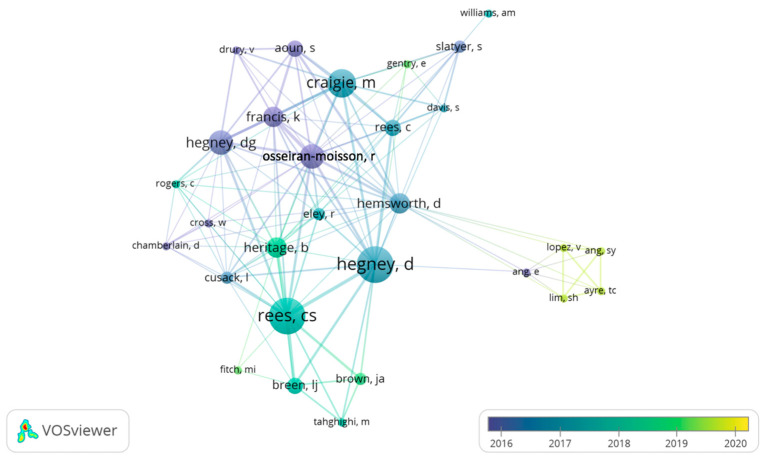
Overlay network and clusters of authorship.

**Figure 6 jpm-12-01574-f006:**
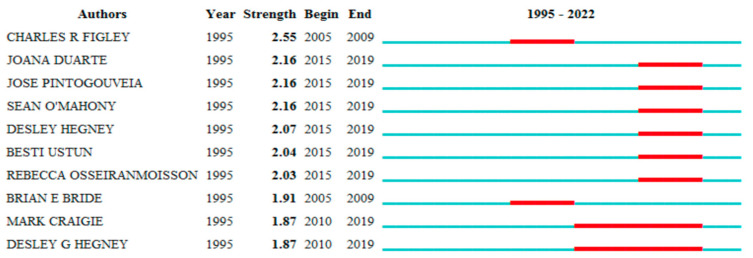
Top 10 authors with the strongest citation bursts.

**Figure 7 jpm-12-01574-f007:**
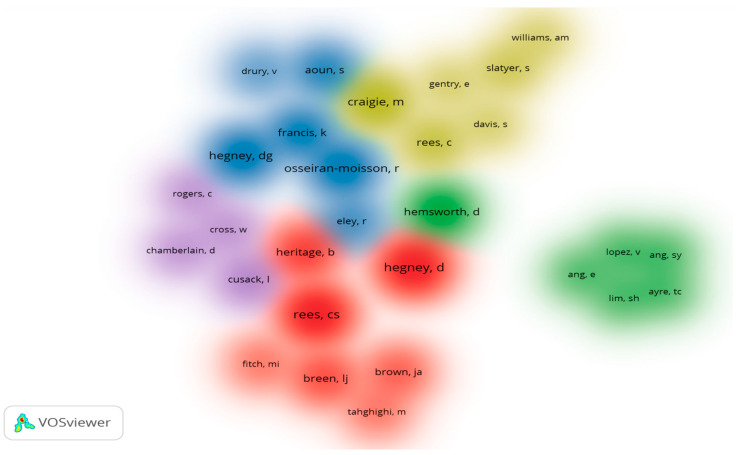
Density of authors’ co-authorship.

**Figure 8 jpm-12-01574-f008:**
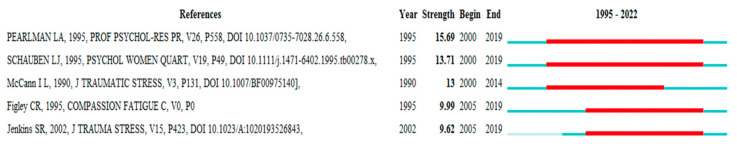
Top 5 references with the strongest citation bursts.

**Figure 9 jpm-12-01574-f009:**
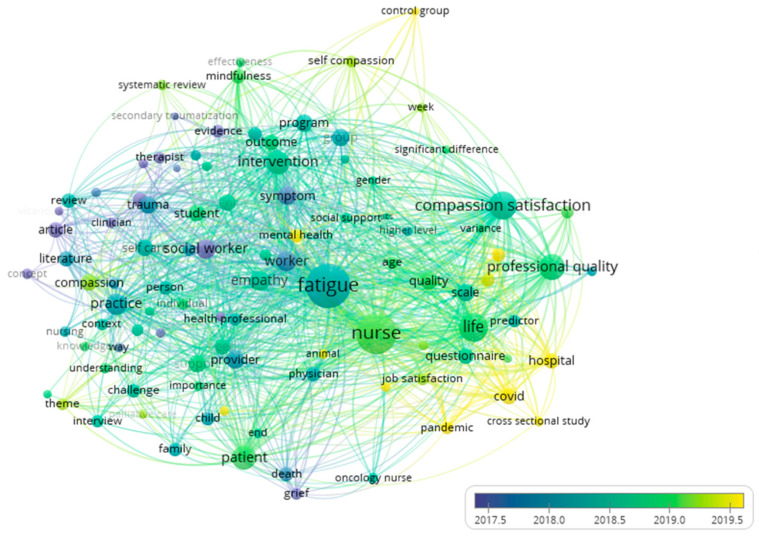
Overlay network of keywords’ co-occurrence.

**Figure 10 jpm-12-01574-f010:**
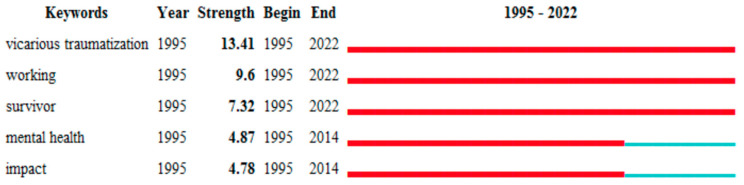
Top 5 keywords with the strongest citation bursts.

**Table 1 jpm-12-01574-t001:** Top 5 Journals with highest impact factor.

Publication Title	Area	Scimago Journal Rank (Quartiles) *	Journal Citation Reports **
*Journal of Nursing Management*	Leadership and Management	Q1 SJR 2021—1.11	3.325
*Frontiers in Psychology*	Psychology (miscellaneous)	Q1 SJR 2021—0.87	2.988
*Journal of Clinical Nursing*	Nursing (miscellaneous)	Q1 SJR 2021—0.83	3.036
*International Journal of Environmental Research*	Health, Toxicology, and Mutagenesis	Q1 SJR 2021—0.81	3.390
*Journal of Advance Nursing*	Nursing (miscellaneous)	Q1 SJR 2021—0.77	3.187

* https://www.scimagojr.com/journalrank.php (accessed on 1 September 2022); ** https://jcr.clarivate.com/jcr/home. (accessed on 1 September 2022).

## Data Availability

Not applicable.

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
