# Peer review of "Bibliometric Analysis of the Scientific Production on Compassion Fatigue"

_jpm, 2022, doi:10.3390/jpm12101574_

Round 1

Reviewer 1 Report

Dear Authors, 

Greetings to all. It's a well-written article, and that will be helpful to many readers. It's a great attempt. A few observations have been made that can be rectified for better enhancement of the article. 

1. Abstract shall be rewritten,  it's not clear.

2. Methodological flowchart can be given for better understanding. 

3. Outcomes of the article are not explained well enough. that should be considered very important while writing a bibliometric paper. 

4. Research objectives are not clearly mentioned in the paper. 

5. It's not much discussed the various fatigues. That shall be the vital focal area rather than others. 

6. Numbers of the papers reviewed?

7. Did you check proper quality checking for the papers? If means whats that method? 

Kindly rectify the given observations to intensify the quality of the paper. best wishes. 

Author Response

The questions you raise, for which we thank you in advance, are extremely pertinent and will be considered, namely at a later stage of the research on the subject, which is underway.

We thank you for your valuable feedback and are willing to change whatever is necessary to improve the article.

  1. Concerning the abstract, we made a brief introduction about the theme, exposing its problematic. The objective was enunciated in a clear and succinct way, having clarified the method to which we resorted. The main results (research frontiers) are enumerated, having clarified the conclusion.
  2. This research method does not require a methodological flowchart as it is not a review study, but an original study.
  3. In the scope of the outcomes, we intend to expose the bibliometric indicators identified, through tables and outputs illustrating the analysis programs themselves, highlighting the research frontiers. We can visualize this type of approach issued by Donthu et al, on which this article was based at the methodological level, making them available for consultation if necessary: https://doi.org/10.1016/j.jbusres.2021.04.070
  4. Regarding the research objectives, these are mentioned in lines 81 to 85, being exposed the research question in the following paragraph.
  5. We understand the relevance of the observation, however the objective of this paper will be to describe compassion fatigue as well as the scientific production associated with it and indexed in the Web of Science to the detriment of any other type of fatigue.
  6. Since this is not a systematic review, we do not present a flowchart where we mention the number of articles reviewed, nor do we present tools to evaluate the quality of the articles. The guidelines by which this document is oriented (Donthu et al) do not contemplate these methodological steps.
  7. We refer you to the answer to the previous point.

Once again, thank you for the learning opportunity, we appreciate the revision made and we remain available to make any changes you find crucial for the improvement of the article.

Regards

The team of authors

Reviewer 2 Report

The authors wrote: "Regarding evidence selection criteria, we established the inclusion of reviews and empirical studies published in scientific journals in article format, addressing or referring to the concept of compassion fatigue, with no chronological limit of publication, and published in any language."  and then:  

„..... Results  - From a sample of 1364 published articles obtained since 1995".....

„The timeframe was established based on the first publication on the topic (1995) to the present (2022)“.

·     But was the first study in 1995 for sure?

 It is an interesting and valuable analysis. It gives information about the amount, type and place of research. we also get to know the main researchers. However in the future, it is worth doing  a systematic review with meta-analysis on the interventions that have the best results in reducing compassion fatigue.

·      There are no real conclusions for nursing practice - they are too general.

Author Response

We appreciate your valuable feedback, will definitely increase value to the document.

With regrads to the questions raised we would like to reply:

Based on the Web of Science database, and as described in the article (figure 1), the first study published on the subject dates back to 1995, and from that date to the date of data collection (2022), there were a total of 1364 articles published on the subject.

Regarding the development of systematic reviews with meta-analysis, it was something we have already addressed in sub-topic 4.4, specifically line 380 and 381 where it states: “a systematic review with meta-analysis on the interventions that have the best results in reducing compassion fatigue".

Regarding the inclusion of implications for nursing practice, the paragraph beginning on line 354 has been added which states, "As nurses represent the largest number of elements of the health care system and given their functional content, they have the highest rates of compassion fatigue."

The purpose of this study was to collect information on the scientific production in this conceptual area. The identification of the authors with the highest scientific production is essential to identify the sources of knowledge and create contact points with experts in the area when developing intervention programs on compassion fatigue.

Once again, we appreciate you feedback and we are remain willing to make the changes you may find crucial.

Regards

The team of authors

Round 2

Reviewer 1 Report

Dear Authors, 

Good day to you. Nice and good work.